# Phage-Mediated Molecular Detection (PMMD): A Novel Rapid Method for Phage-Specific Bacterial Detection

**DOI:** 10.3390/v12040435

**Published:** 2020-04-11

**Authors:** Francisco Malagon, Luis A. Estrella, Michael G. Stockelman, Theron Hamilton, Nimfa Teneza-Mora, Biswajit Biswas

**Affiliations:** 1Genomics & Bioinformatics Department, Biological Defense Research Directorate, Naval Medical Research Center-Frederick, Fort Detrick, MD 21702, USA; luis.a.estrella.mil@mail.mil (L.A.E.); theron.c.hamilton.mil@mail.mil (T.H.); 2Wound Infections Department, Infectious Diseases Directorate, Naval Medical Research Center-Silver Spring, Silver Spring, MD 20910, USA; michael.g.stockelman.mil@mail.mil (M.G.S.); nimfa.c.tenezamora.mil@mail.mil (N.T.-M.); 3Henry M. Jackson Foundation for the Advancement of Military Medicine, Bethesda, MD 20817, USA; 4Leidos, Reston, VA 20190, USA

**Keywords:** bacteriophage, phage, RNA, bacterial detection, antibiotic resistance, *S. aureus*, *B. anthracis*

## Abstract

Bacterial infections pose a challenge to human health and burden the health care system, especially with the spread of antibiotic-resistant populations. To provide effective treatment and improved prognosis, effective diagnostic methods are of great importance. Here we present phage-mediated molecular detection (PMMD) as a novel molecular method for the detection and assessment of bacterial antibiotic resistance. This technique consists of a brief incubation, of approximately ten minutes, of the biological sample with a natural bacteriophage (phage) targeting the bacteria of interest. This is followed by total RNA extraction and RT-PCR. We applied this approach to *Staphylococcus aureus* (SA), a major causative agent of human bacterial infections. PMMD demonstrated a high sensitivity, rapid implementation, and specificity dependent on the phage host range. Moreover, due to the dependence of the signal on the physiological state of the bacteria, PMMD can discriminate methicillin-sensitive from methicillin-resistant SA (MSSA vs. MRSA). Finally, we extended this method to the detection and antibiotic sensitivity determination of other bacteria by proving PMMD efficacy for *Bacillus anthracis*.

## 1. Introduction

*Staphylococcus aureus* (SA), a gram-positive bacteria, causes human infections, a fact that has been long-recognized [1] and is still the case today [2]. The use of antibiotics was initially very effective against SA infections, but evolution and selection has led to the enrichment of antibiotic-resistant SA populations [3,4]. Molecular detection by PCR using primers against the SA genome has been previously reported [5], and further refined [6,7]. Nevertheless, this sensitive technique does not directly assess the growth inhibition of the cells by antibiotics but rather, the genomic makeup of the bacteria [5,7]. These are important reasons why clinical testing is still carried out using lengthier physiological assays.

Phages refers to a group of extremely diverse and abundant viruses that infect bacteria [8]. Phage infection is highly efficient and host specific, a fact used by researchers for the delivery of cargo into bacteria [9,10,11]. In the case of lytic DNA phages, the injected genomic DNA molecule directs the very rapid production of a high number of progeny phages per infected cell, typically dubbed as the burst size. This is the base for phage-typing bacterial diagnostic plaque assays, further refined using labeled phages, anti-phage antibodies, or genetically engineered reporter-containing phages [12]. The dependence of phage life cycle progression on the physiological state of the bacteria also allows for the evaluation of antibiotic resistance using phage immunodetection methods [13]. Nevertheless, although robust, this antibiotic susceptibility assay requires the production of good antibodies and the incubation of the biological sample in BacT/ALERT culture media for SA amplification and identification [14].

Compared with antibody-based methods, PCR-based nucleic acid approaches have significantly better LOD (limit of detection) and are not constrained by antibody availability. Moreover, detection methods based on the reverse transcription of highly abundant RNAs show better LOD and are less prone to false positives than DNA detection assays [15]. Interestingly, Mulvey et al. [10,16] reported the use of TM4 genetically engineered phages for the detection and antibiotic susceptibility determination of *Mycobacterium tuberculosis*. In this assay, the phage delivers heterologous cassette(s), “surrogate marker loci” (SML), designed to produce high RNA levels. Upon incubation with the biological sample for several hours to allow for phage/SML amplification the cassette expression is detected by RT-PCR. We considered applying the SML approach for SA, but the introduction of heterologous markers in phage vectors depends on the idiosyncrasy of each viral genome. In the case of phages not previously characterized for this purpose, this could be a cumbersome and slow process. Indeed, similarly to “eukaryotic viruses”, the word “phages” encompasses a plethora of organisms [17,18]. Thus, instead of introducing SMLs into SA phages, we decided to use natural phage RNAs as markers for SA detection and antibiotic testing. In addition, we reasoned that, rather than relying on phage particle multiplication, the rapid viral transcription peak subsequent to phage genome injection should suffice to fulfill our goals.

Here, we report a novel molecular detection method for SA. This method, dubbed phage-mediated molecular detection (PMMD), consists of a short incubation of the culture with a SA-specific phage followed by RNA extraction and RT-PCR. We demonstrate that the spurt of viral metabolism of natural phages shortly after infection is enough for the production of a strong signal. This method reduces culture incubation time, is highly specific and sensitive, and can be applied to complex biological tissues, i.e., without the need to isolate SA from the clinical sample. Finally, it allows antibiotic susceptibility determination.

## 2. Materials and Methods

Unless indicated otherwise, the methods involving commercial kits were performed following manufacturers’ instructions.

### 2.1. Growth and Phage Assay Conditions

Bacterial cultures were grown for one hour in LB-Lennox (10 g/L tryptone, 5 g/L yeast extract, and 5 g/L NaCl) media at 37 °C with strong aeration (225 rpm shaking). For antibiotic sensitivity assays, the cultures were grown for one or three hours, as indicated. Cefoxitin (Sigma-Aldrich, St. Louis, MO, USA) and tetracycline (Sigma-Aldrich) were added at final concentrations of 4 µg/mL, using a 1000× stock in water, and 5 µg/mL, using a 200× stock in water, respectively.

The reactions were performed by mixing 50 µL of bacterial cultures (~2 × 10^9^ cfu/mL) and 50 µL of a phage K lysate (~5 × 10^8^ pfu/mL) and 1 µL of CsCl gradient purified phage K (~ 1 × 10^10^ pfu/mL) or 1 µL of CsCl gradient purified phage Gamma (~1 × 10^10^ pfu/mL). The mixtures were incubated on ice for 3–4 minutes and the infections were allowed to proceed for 10 minutes, unless indicated otherwise, at 37 °C with shaking (400 rpm). For SA, five minutes prior to the end of the incubation, lysostaphin (Sigma-Aldrich) was added from a 20× stock (2 mg/mL).

### 2.2. Phage Lysates and CsCl Gradient Purification

Phage K and phage Gamma lysates were produced by infecting *S. aureus* K1 and *B. anthracis* Sterne, respectively, following standard procedures. The lysates were further processed by centrifugation at 3500 *g* for 10 minutes and filtering the supernatant using 0.22 µm pore Steriflip filter units (Millipore, Burlington, MA, USA). Phage purification using CsCl gradients was performed as previously described [19]. Briefly, the four liters of host strain was grown to 0.1 OD_600_ at 37 °C and infected with phage at a multiplicity of infection of ~0.5, and incubated at 37 °C until lysis occurred (as determined by a sharp decrease on OD_600_). The lysate was cleared via centrifugation at 10,000 *g* for 10 min and the phages precipitated by the addition of 10% *w*/*v* of polyethylene glycol 8000 and overnight incubation at 4 °C. The solution was centrifuged at 5000 *g* for 1 h, the supernatant decanted, and the pellet resuspended in 5 mL of SM buffer. Then, 0.75 g of cesium chloride per ml of precipitate was added, mixed by inversion, and centrifuged on a 90 Ti rotor at 58,000 *g* at 4 °C for 24 h. The resulting band was retrieved and dialyzed by using a 10,000 Da MWC Slide-A-Lyzer dialysis cassette in 4 L of dialysis buffer (100 mM NaCl, 8 mM MgSO_4_, and 50 mM Tris-HCl).

### 2.3. RNA Preps, DNA Preps, Gel Electrophoresis and Sequencing

SA RNA preparations were performed using the hot-phenol method [20] followed by treatment with amplification grade DNAse I (Thermo Fisher Scientific, Waltham, MA, USA), or using Direct-zol RNA MicroPrep w/ TRI Reagent kits (Zymoresearch, Irvine, CA, USA). BA RNA preparations were performed by adding 25 µL of “lysis mix” (a mixture of 200 µL of 10% SDS and 50 µL of 0.5 M EDTA), 5-min incubation at 99 °C, followed by the hot-phenol method [20]. Total bacterial RNAs amount and integrity was QCed (Quality Controlled) by absorbance readings using a Nanodrop spectrophotometer and by observation of rRNAs on 0.8% agarose gels stained with ethidium bromide. Phage K DNA preparations were made by proteinase K treatment followed by phenol/chloroform extraction as indicated previously [19]. Nucleic acid integrity was evaluated by gel electrophoresis in ethidium bromide agarose gels. The1 kb plus DNA ladder (Thermo Fisher Scientific) was used as a size marker. DNA band isolation was done using the Purelink quick gel extraction kit (Thermo Fisher Scientific). Sanger sequencing was done by MacrogenUSA using the oligos OK1-F and SeqI2-F, and the resulting peaks were visualized using opensource ApE plasmid editor software [21].

### 2.4. PCR and qPCR Reactions and Analysis

Regular PCRs were done using Phusion High-Fidelity DNA polymerase (New England Biolabs, Ipswich, MA, USA) and a C1000 thermal cycler (BioRad, Hercules, CA, USA). Stepwise RT-PCRs were done using the SuperScriptIII First-Strand Synthesis System (Themo Fisher Scientific) and the oligo OK1-R for priming, followed by thermal amplification with Phusion High-Fidelity PCR kit (New England Biolabs). The PCR conditions using the oligos OK1-F and -R were: 98 °C 2 min/ 35× (98 °C, 30 s; 50 °C, 30 s; and 72 °C, 6 min)/ 72 °C, 10 min. When using primer pairs OK2-F and -R or OK3-F and -R, the elongation time was reduced to 1 min.

Single-step fluorescent qPCR was done with Luna one-step RT-qPCR kit (New England Biolabs) using low profile clear 0.2 mL PCR tubes and optical ultraclear flat caps (BioRad). The reactions were conducted in a CFX96 real-time PCR detection system (BioRad) with CFX Manager 3. Cqs were auto calculated by the software, and ranged from ~190 to ~250. The single-step RT-qPCR conditions were as follows: 55 °C, 10 min/35× (95 °C, 10 s and 60 °C, 30 s, plate read).

### 2.5. Oligonucleotides

All the primers were designed manually based on the Genebank phage K sequence NC_005880 [22,23], or phage Gamma (isolate d’Herelle) sequence DQ289556.1 [24], and synthesized by GeneOracle. The oligos used were: OK1-F (GAGTTGGTAGATAACATTG), OK1-R (AATTGTTCATCTGTTAGTTTACCTGACTCTTTATAATC), SeqI2-F (CTACAATGAGAGAGCACTGG), OK2-F (GAATTGCTTAAGAATTGGTTAGCTAG), OK2-R (CCTGAATACTCTCAAAATCTTTAAAG), OK3-F (ACAAGAAAGCTATTGGTTATGCGTTAGATAATTTAG), OK3-R (TGTAGCATATCAGGGTCTTTAGTAAATAATCCGA), Gamma-pol-f (GTATACATTGCAGGGATTC-AAG) and Gamma-pol-rev (CATGTTCGTCATTTAATTTCTCACC).

### 2.6. Bacterial Strains and Phages

Phage K [25] and the following bacterial strains were obtained from ATCC: *S. aureus* strain K1 [26], *Staphylococcus epidermidis* Fussel strain [27], and *Pseudomonas aeruginosa* strain O1 [28]. Phage Gamma [29,30] and the following bacteria are maintained in our lab collection: *S. aureus* strain RN4220 strain [31], *Staphylococcus haemolyticus*, *S. aureus* strains SA1415.COI, SA1012.COI, SA1107.COI, and SA1019.COI [19], *Acinetobacter baumannii* strain AB5075 [32], and *B. anthracis* Sterne (strain 7702) [33].

## 3. Results

### 3.1. Detection of SA by Phage K RT-PCR Plus Gel Electrophoresis

In order to detect SA, we took advantage of the high metabolic rate of phages during infection. We thus designed an approach (PMMD) for SA consisting of the infection of the bacteria with phage K followed by detection of phage RNA, as illustrated in Figure 1A. Briefly, the SA sample is incubated with phage K, high quality RNA is then extracted, reverse-transcribed, and subjected to PCR amplification. We chose gp125 RNA as the amplification target (Figure 1B). This RNA encodes for a putative DNA polymerase and contains two group I introns [19]. We reasoned that targeting an intron-containing gene could provide a quality control point for the RT-PCR by analyzing the product size. The oligo OK1-R was designed for RT priming, a step to be followed by PCR amplification with the oligos OK1-F and OK1-R.

Under these conditions, we expected the amplification of a major 2.6 kb band, corresponding to the spliced RNA (mRNA), in the samples where productive infection had occurred (Figure 1C). In addition, minor upper size band(s), corresponding to unspliced (pre-mRNA) and/or incorrectly spliced RNA, may also appear, as observed in Figure 2. In contrast, regular PCR of genomic phage K DNA should produce a single band of 4.6 kb. The empirical application of the assay with the SA RN4220 strain, as described in Material and Methods, fully agreed with our expectation (Figure 2A), and revealed that the optimal incubation time is ~10 minutes. The partial sequencing of the 2.6 kb product with oligos specific for the area around the introns further confirmed that the diagnostic band results from the splicing of introns 1 and 2, and agreed with the predicted exon junctions at the nucleotide level (Figure 2B and [22]).

### 3.2. Phage Host-Range Specific Bacterial Detection

PMMD should be dependent on the specificity of the phage. Therefore, we tested a set of four bacterial isolates belonging to the USA300 series, the most common community-associated methicillin-resistant group in the USA [34], previously characterized as phage K-sensitive in our lab [19]. As shown in Figure 2C, all bacterial isolates tested positive with our assay. In addition, it was important to test the specificity and influence of the presence of additional bacteria in the culture. Therefore, we challenged the method with four bacteria strains where phage K cannot propagate, as tested by the double agar method (*Staphylococcus epidermidis* (SE), *Staphylococcus haemolyticus* (SH), *Acinetobacter baumannii* (AB), and *Pseudomonas aeruginosa* (PA); see Materials and Methods). As expected, in the cases tested here as proof of concept, PMMD-SA did not produce positive signals for non-SA, and was able to detect SA in complex mixtures (Figure 2D).

### 3.3. LOD Using Fluorescent Quantitative RT-PCR

PCR analysis by gel electrophoresis is, at best, only semiquantitative and, although easy to perform and inexpensive, is not the method of choice for most clinical laboratories. To improve the workflow by analyzing the RT samples using fluorescent qPCR, we decreased the size of the diagnostic amplicon, a necessary step for qPCR assays. We then repeated the assay using oligos specific for intron 1 or intron 2 splicing events (Figure 3A,B), obtaining the expected results in both cases. Analysis of single-step fluorescent qPCR of the same amplicons (Figure 3C) indicated that both primer pairs effectively produce diagnostic signals, and that the signal specific for intron 1 splicing is stronger. Thus, we decided to use this primer pair for subsequent experiments. Next, we confirmed the specificity results using this new assay design. Mixed cultures of different uninfected bacteria, as in Figure 2D, did not produce false positives. On the other hand, mixed cultures incubated with phage K only showed a prominent signal in the presence of SA (Figure 4A). This validated the RT-qPCR approach for detection of SA in complex samples.

With the current assay design, involving hot-phenol acid RNA extraction and the use of an unpurified lysate of phage K, a strong signal is produced with ~10,000 bacteria (data not shown). We increased the LOD using a column-based RNA purification method (Direct Zol RNA prep Kit, see Material and Methods) (Figure 4B). The use of this kit not only improves the LOD but also significantly simplifies and speeds up the RNA extraction step. Due to signal production by unpurified lysates after cycle 35, to increase the number of cycles to 40 without false positives we used CsCl gradient purified phage K for the assay. Under this condition of PMMD application to cultures of SA diluted at different degrees, the assay sensitivity was ~100 CFU (Figure 4B), as determined by colony forming units counting.

### 3.4. Discrimination of MSSA Versus MRSA

Small adjustments of PMMD should allow this technique to be used for applications other than detection. Since the initial step requires an incubation time of ~1 hour for the bacteria to increase their metabolic rate, we reasoned that the assay could differentiate between MSSA and MRSA if the cultures were incubated with and without cefoxitin (fox), an antibiotic used for susceptibility testing of MRSA isolates [35,36]. To this end, we used MIC (minimal inhibitory concentration) of fox, this is, the lowest concentration which prevents visible bacterial growth after 24 h incubation. This implies that the concentration of antibiotic is sufficient to inhibit the metabolic level of the cell required for growth after one day, but is not at a concentration which is high enough to completely shut down bacterial metabolism. As shown in Figure 5A, the addition of fox in the media increased the Cq values in the MSSA strain RN4220 by ~2.0 and ~10.5 after 1 and 3 hours of incubation with the antibiotic, respectively. By contrast, for the MRSA strain SA-1012.COI, the presence of fox had a minimal effect on Cq under the same conditions. Since a variation of Cq of 3.3 to 4.0, depending on the amplicon, corresponds to a difference of 10-fold in signal, PMMD seems to be extremely sensitive at MIC. Even after only 3 hours, the MSSA strain showed a variation of Cq of ~10.5, corresponding roughly to a higher than 100-fold decrease on the levels of phage RNA, with practically no change in the case of the MRSA strain. We thus conclude that the comparison of PMMD results after incubation with and without fox prior to infection can be used for the determination of methicillin resistance.

### 3.5. PMMD Applied to Bacillus Anthracis

To demonstrate broader applicability of the method for detection and antibiotic sensitivity of other bacteria, we followed a similar approach for *B. anthracis* (BA), with phage Gamma. Thus, we designed oligos for the amplification of a 118 bp fragment specific for the dnaC region of Gamma. In addition to detection, we also wanted to test antibiotic sensitivity, in this case to tetracycline. The molecular protocol was similar to the one used for SA. As can be observed in Figure 6, we are able to efficiently apply the PMMD method for BA detection.

## 4. Discussion

Because of the inherent nature of phage specificity to their host, phage diagnostics can promptly and sensitively detect their specific host in a variety of clinical and environmental samples. Certainly, a number of previous studies have reported the successful use of phage PCR methods for bacterial detection e.g., [37,38,39]. Nevertheless, the inherent background signal associated with the input requires the propagation of the phages to produce a progeny level sufficient to overcome this issue. Therefore, upon infection, relatively long incubation times are need it. This was cleverly resolved by Mulvey and collaborators by focusing on the detection of RNA molecules synthetized by a highly-transcribed phage-borne artificial cassette instead of detecting phage DNA [10,16]. Due to the fact that the number of phage RNA molecules produced per infected cell generally far exceeds the number of new phage DNA genomes, we explored the possibility of developing phage RNA detection as a reliable and rapid diagnostic tool.

The results presented support PMMD as an alternative technique for medical diagnosis of *S. aureus*. First, PMMD discriminates between antibiotic sensitive and resistant bacteria. As proof of principle, we have used cefoxitin, the CDC (Centers for Disease Control and Prevention)- and CLSI (Clinical & Laboratory Standards Institute)-recommended drug for testing of MRSA strains. Although the use with other antibiotics has to be determined empirically, we cannot anticipate major problems in adapting the system for other drugs, either bactericidal or bacteriostatic, since the active growth of the bacteria is essential for the phage metabolism [40]. However, firstly compared with PCR approaches that amplify gene resistance markers, PMMD can be used for a number of antibiotics, does not require different primer pairs for different resistance cassettes, and avoids false positives due to the presence of silent or mutant cassettes [40]. Secondly, the LOD to obtain a strong signal is ~100 bacterial cells. It is important to note that although the background signal was almost null for the uninfected bacteria controls, there was a small amount of background signal which interfered with our LOD results when unpurified phage lysates were used. Thus, highly-purified phages, in our case obtained by a CsCl-gradient isolation procedure, are an essential requirement for high sensitivity. Thirdly, the method is fast; we estimate its implementation time to be ~ 3 hours or ~5 hours depending on whether only detection or detection plus antibiotic sensitivity is performed. This time frame could be decreased by automation of the process and adjustment of the incubation periods depending on the antibiotic and culture media used. For example, we use 1-hour incubation time in LB media to promote high bacterial metabolic rate, but it is known that significantly shorter incubation times also work [25]. Finally, PMMD is specific and can be applied to complex mixtures. The specificity level is determined by the phage used. Phage K infects the majority of *S. aureus* strains and only propagates in a few isolates of closely-related *Staphylococcus*. This is the case of *Staphylococcus hyicus*, used sometimes as a surrogate for phage K propagation for safety reasons [19,41]. Moreover, co-infection with other phage(s) can raise the number of positive SA strains to >95% [18,19,41]. In addition, due to the simplicity of PMMD, there is no need to genetically engineer the phage; this approach can be easily expanded to other bacterial species, as we exemplified here in the case of *B. anthracis*.

In summary, we have developed PMMD as a novel, proof-of-concept method for the detection and drug sensitivity assessment of bacteria using an *S. aureus*/phage K pair. PMMD shows promising characteristics for its applicability for clinical and research diagnostic purposes. Nevertheless, further development is required to cover most bacterial strains of a given species, for example by using phage cocktails. Moreover, this methodology has the potential to be adapted for the rapid selection of phages active against clinical isolates during phage therapy.

## 5. Patents

Malagon, F., Estrella, L.A., and B. Biswas. 2018. Phage-Mediated Molecular Detection Methods and Related Aspects. USSN 15/994,855 and PCT/US18/35449. https://patentscope.wipo.int/search/en/detail.jsf?docId=WO2018222907.

## Figures and Tables

**Figure 1 viruses-12-00435-f001:**
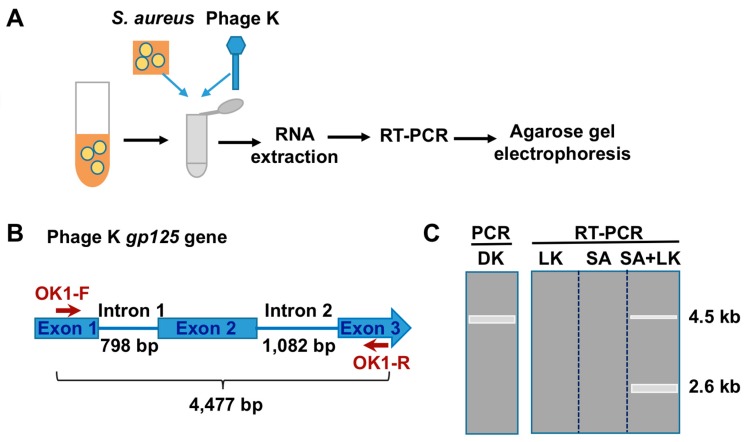
Schematic representation of the method and the predicted results using oligos in the 5’ and 3’ ends of phage K gp125 ORF. (**A**) Schematic representation of the method (see text for details). (**B**) Region and size of phage K gp125 amplified by the primers OK1-F and -R. The sizes of introns 1 and 2 are also indicated. (**C**) Expected gel electrophoresis results of PCR from phage K genomic DNA (DK), and from reverse-transcribed phage K lysates (LK), *Staphylococcus aureus* cells (SA), and SA cells infected with phage K (SA+LK). The predicted bands (light grey rectangles) and their sizes are indicated.

**Figure 2 viruses-12-00435-f002:**
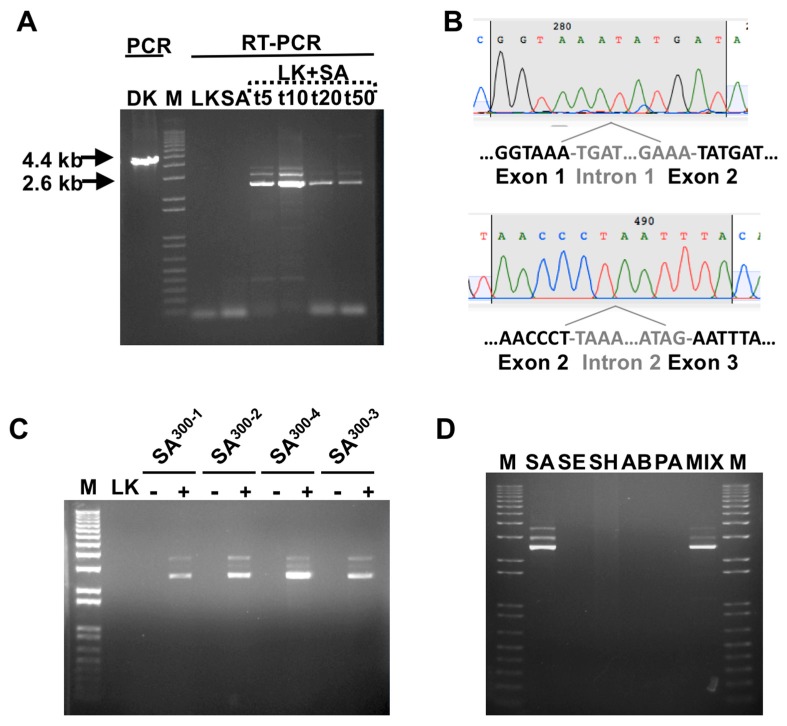
Page-mediated molecular detection (PMMD) assay by two-step RT-PCR. (**A**) Gel electrophoresis results of PCR from reverse-transcribed phage K lysates (LK), SA cells (SA), and SA cells infected with phage K (SA+LK) for 5, 10, 20, and 50 minutes. PCR from phage K genomic DNA (DK) and a commercial size marker (M) are also shown. The approximate sizes of the major bands are indicated with arrows. Minor upper size band(s), correspond to unspliced (pre-mRNA) and/or incorrectly spliced RNAs. (**B**) Sanger sequencing chromatograms of the 2.6 kb RT-PCR band shown in (**A**) using primers specific for the predicted exon–exon junctions. The exon junction (black font) and intervening introns (gray font) sequences are also shown. (**C**) Gel electrophoresis of PMMD results, + lanes, using four USA300 strains (SA300-1 to -4, corresponding to clinical isolates SA1415.COI, SA1012.COI, SA1107.COI, and SA1019.COI, respectively). Controls without infection, - lanes, and with phage K lysate without cells (LK), as well as a DNA size marker (M), are also shown. (**D**) Gel electrophoresis of PMMD results from different bacteria incubated with phage K (SA: *Staphylococcus aureus*, SE: *Staphylococcus epidermidis*, SH: *Staphylococcus haemolyticus*, AB: *Acinetobacter baumannii*, and PA: *Pseudomonas aeruginosa*, MIX: mixture of all the bacterial species), and M: DNA size marker.

**Figure 3 viruses-12-00435-f003:**
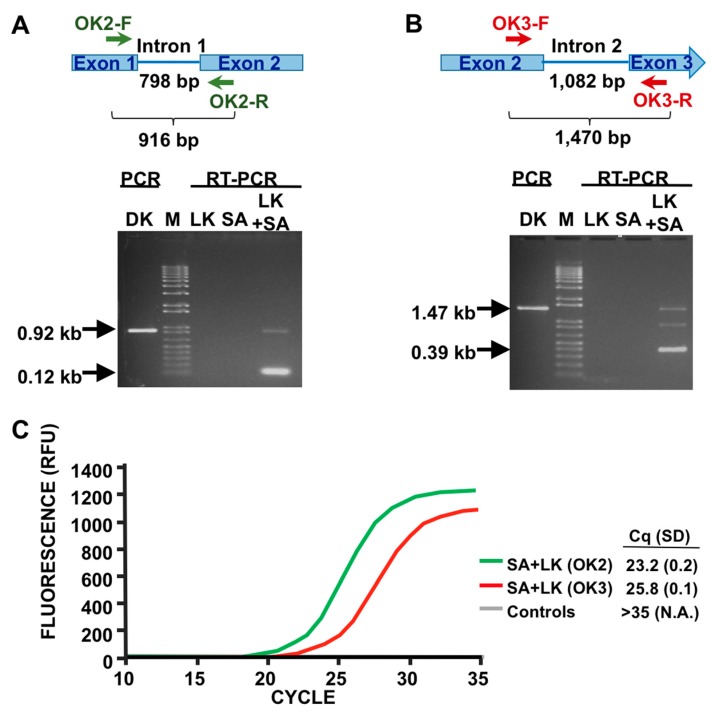
PMMD) with primer pairs OK2 and OK3 and fluorescent qPCR comparison. (**A**) Region and size of phage K gp125 amplified by the primers OK2-F and –R (the size of intron 1 is also indicated); and gel electrophoresis results of PCR from reverse-transcribed phage K lysates (LK), SA cells (SA), and SA cells infected with phage K (SA+LK). PCR from phage K genomic DNA (DK) and a commercial size marker (M) are also shown. The approximate sizes of the major bands are indicated with arrows. (**B**) Region phage K gp125 amplified by the primers OK3-F and -R and gel electrophoresis. Scheme details as in panel (A). (**C**) Fluorescent qPCR of RNA samples amplified with the primer pairs OK2 and OK3. The infections were conducted with diluted SA samples (10^−4^) to avoid signal saturation. The colored lines’ correspondence, relative fluorescent units (RFU), number of cycles and Cq average, and S.D. values of three biological replicates performed in parallel are shown. As indicated, the control templates (lysate K, SA cells, and water) did not produce signal either with primer pair OK2 or with OK3.

**Figure 4 viruses-12-00435-f004:**
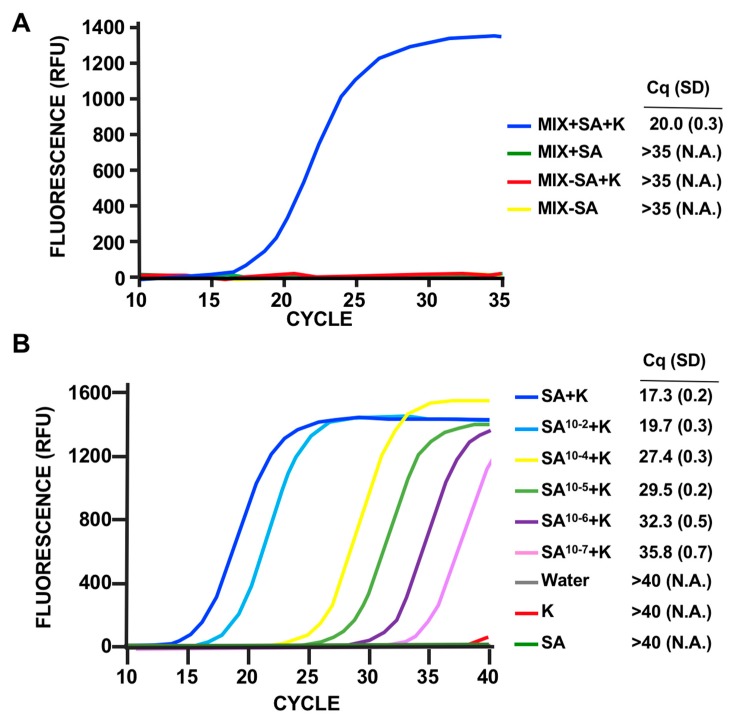
Specificity and sensitivity of PMMD by fluorescent qPCR using the OK2 primer pair and CsCl gradient purified phage K. (**A**) Specificity of the assay using mixed bacterial populations with and without SA. The bacterial species used as in Figure 2D (see text for details). Representation of the results as in Figure 3C. (**B**) Sensitivity of the assay of SA samples diluted as indicated. The number of bacteria ranged from 10^9^, for the undiluted sample, to 100 cells in the higher dilution (i.e., 10^−7^), as determined by CFU counting. Representation of the results as in Figure 3C.

**Figure 5 viruses-12-00435-f005:**
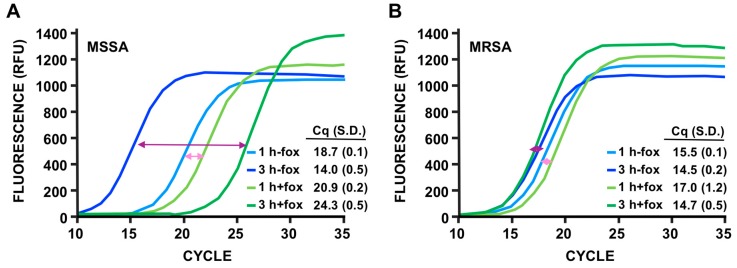
Discrimination of methicillin-sensitive SA (MSSA) versus methicillin-resistant SA (MRSA). (**A**) Fluorescent qPCR results of PMMD of the MSSA strain RN4220. The cultures were incubated with or without cefoxitin (fox) for 1 or 3 hours, as indicated. Representation of the results as in Figure 3C. Uninfected cultures, purified phage K and water controls did not produce signal. (**B**) Fluorescent qPCR results of PMMD of the MRSA strain SA1012.COI. The cultures were incubated with or without fox for 1 or 3 hours as indicated. The colored lines’ correspondence, relative fluorescent units (RFU), number of cycles and Cq average, and S.D. values of three biological replicates performed in parallel are shown. The double-headed arrows were added to better identify the curves comparing conditions with and without fox after 1 hour (light pink) and 3 hours (dark pink). Uninfected cultures, purified phage K, and water controls did not produce signal.

**Figure 6 viruses-12-00435-f006:**
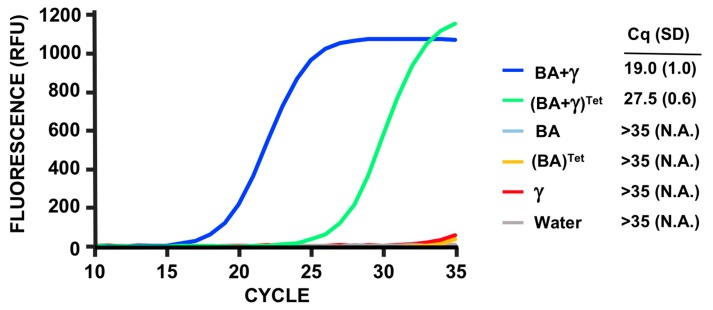
Fluorescent qPCR results of *B. anthracis* Sterne detected using phage Gamma RNA production. RNA levels from phage Gamma (γ), BA cells (BA), and BA cells infected with phage Gamma for 10 minutes (BA+ γ) were used for the assay (see text for details). Bacterial cultures grown in the presence of tetracycline (Tet) are indicated. The colored lines’ correspondence, relative fluorescent units (RFU), number of cycles, and Cq average and S.D. values of three biological replicates done in parallel are shown.

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
