# Peer review of "Phage-Mediated Molecular Detection (PMMD): A Novel Rapid Method for Phage-Specific Bacterial Detection"

_viruses, 2020, doi:10.3390/v12040435_

Round 1
Reviewer 1 Report
This is interesting technology for rapid and sensitive detection of bacteria and for the identification of antibiotic susceptibility. The paper is also well written.
Minor editing issues/suggestions:
Text is no longer formatted as ‘Justified’ after line 74
Line 196: change “all of them” to “all bacterial isolates tested”
Line198 change “fours” to “four”
Line 201: change “produces no false positive signal” to does not produce positive signals”
Line 213: change “do not” to did not” or “does not”
Line 214 change “result in a prominent signal…” to only showed a prominent signal in the presence of SA”
Figure 3. Hard to see text of Exon 1 and Exon 2. Change to a darker text and/or lighten shading of blue boxes
Arrows (green and red) need more spacing from text for better visualization
Line224: change (10-4) to (at 10-(4)) or use superscript for -4
Figure 4B: Hard to differentiate between brown, orange and red colors. Suggest different color choice.
Line 323: change “(also see text for details)” to “(see text for details)”
Line 235: Consider adding (i.e. 10-7)
Line237: remove the word “reliably”
Line 237: change to “(data not shown)”
Line 238: remove the word ‘by’
Line 241-243: It is noteworthy – sentence structure issues
Line 252: suggest not shortening cefoxitin to fox throughout the text as author switches back and forth between the two names (refer to line 251, 255 ,257
Line 252: suggest the following “This implies that the concentration of antibiotic is sufficient to inhibit the metabolic level of the cell required for growth after one day, but is not at a concentration which is high enough to completely shut down bacterial metabolism. “
Line 256: elsewhere author uses 1 and 3 hours
Line 258: add “s” to “correspond”
Line 259: “MSSA strain”
Line 260: > 100 fold
Line 262: used for the
Figure 5B: Hard to differentiate between the different pink colors;
Line 268 spacing issue
Line 279: we also wanted to test
Line 283: B. anthracis is not italicized
Line 301: Understanding
Line 302: Due to the inherent
Line 307: reliable and rapid
Line 311: we have used cefoxitin
Line 312: use of contraction “can’t” change to “cannot”
Line 315: “However, firstly compared with”
Line 317: Secondly
Line 318: that although the background signal is almost
Line 319: a small amount of background signal which interferes with our LOD results when unpurified phage lysates were used. Thus,
Line 322: Thirdly,
Line 326 1-hour
Line 327: Finally, PMMD
Line 329: the words “great” and “very” are not needed in the sentence
Line 329-331 Phage K – sentence structure issues
Line 332: strains to >95%
Line 332: In addition, due to the simplicity of PMMD, there
Line 335: as a novel, proof of concept method for the
Line 336: using a S. aureus/phage K pair. PMMD shows promising characteristics
Line 336-343: PMMD – sentence structure issues.
Author Response
Dear Sir/Madam,
Thank you very much for your kind words and for your detailed corrections.
In the revised manuscript, we have incorporated all the changes that you suggested, as indicated below.
Very Respectfully,
The authors.
Point by point responses:
Text is no longer formatted as ‘Justified’ after line 74
This have been corrected. All the text in the manuscript is now justified.
Line 196: change “all of them” to “all bacterial isolates tested”
This has been corrected as requested. Due to changes in formatting, this is currently in Line 193.
Line198 change “fours” to “four”
This has been corrected as requested. Due to changes in formatting, this is currently in Line 195.
Line 201: change “produces no false positive signal” to does not produce positive signals”
This has been corrected as requested. Due to changes in formatting, this is currently in Line 198.
Line 213: change “do not” to did not” or “does not”
This has been corrected to “does not”. Due to changes in formatting, this is currently in Line 210.
Line 214 change “result in a prominent signal...” to only showed a prominent signal in the presence of SA”
This has been corrected as requested. Due to changes in formatting, this is currently in Line 211.
Figure 3. Hard to see text of Exon 1 and Exon 2. Change to a darker text and/or lighten shading of blue boxes
Arrows (green and red) need more spacing from text for better visualization
This has been corrected as requested. The text for Exon 1 and Exon 2 has been changed to a darker color and the blue boxes have been made slightly lighter. There is now more spacing between the green and red arrows and the text.
Line224: change (10-4) to (at 10-(4)) or use superscript for -4
This has been corrected as requested by using superscript for -4. Due to changes in formatting, this is currently in Line 221.
Figure 4B: Hard to differentiate between brown, orange and red colors. Suggest different color choice.
We have kept the red color and have changed brown to green and orange to eggplant.
Line 223: change “(also see text for details)” to “(see text for details)”
This has been corrected as requested. Due to changes in formatting, this is currently in Line 229.
Line 235: Consider adding (i.e. 10-7)
This has been corrected as requested. Due to changes in formatting, this is currently in Line 231.
Line237: remove the word “reliably”
This has been corrected as requested.
Line 237: change to “(data not shown)”
This has been corrected as requested. Due to changes in formatting, this is currently in Line 234.
Line 238: remove the word ‘by’
The word by has been eliminated.
Line 241-243: It is noteworthy – sentence structure issues
The sentence has been changed to “Due to signal production by unpurified lysates after cycle 35, to increase the number of cycles to 40 without false positives we used CsCl gradient purified phage K for the assay”. Now in lines 237 to 239.
Line 252: suggest not shortening cefoxitin to fox throughout the text as author switches back and forth between the two names (refer to line 251, 255 ,257
Cefoxitin has been shortened to fox as requested.
Line 252: suggest the following “This implies that the concentration of antibiotic is sufficient to inhibit the metabolic level of the cell required for growth after one day, but is not at a concentration which is high enough to completely shut down bacterial metabolism. “
This has been corrected as required. Now in lines 248 to 251.
Line 256: elsewhere author uses 1 and 3 hours
This has been corrected by using 1 and 3 hours as requested. Now in line 252.
Line 258: add “s” to “correspond”
This has been corrected. Now in line 255.
Line 259: “MSSA strain”
This has been corrected. Now in line 256.
Line 260: > 100 fold
This has been corrected. Now in line 257.
Line 262: used for the
This has been corrected. Now in line 259.
Figure 5B: Hard to differentiate between the different pink colors;
The color pattern in Figure 5B is now identical to that of Figure 5A.
Line 268 spacing issue
The extra space has been deleted.
Line 279: we also wanted to test
This has been corrected. Now in line 274.
Line 283: B. anthracis is not italicized
We have italicized B. anthracis. Now in line 278.
Line 301: Understanding
This word is no longer present in the manuscript due to suggestion of reviewer 2 to shorten and change the first paragraph of the Discussion section.
Line 302: Due to the inherent
No longer present in the manuscript due to suggestion of reviewer 2 to shorten and change the first paragraph of the Discussion section.
Line 307: reliable and rapid
This has been corrected. Now in line 295.
Line 311: we have used cefoxitin
This has been corrected. Now in line 298.
Line 312: use of contraction “can’t” change to “cannot”
This has been corrected. Now in line 299.
Line 315: “However, firstly compared with”
This has been corrected. Now in line 301.
Line 317: Secondly
This has been corrected. Now in line 304.
Line 318: that although the background signal is almost
This has been corrected. Now in line 305.
Line 319: a small amount of background signal which interferes with our LOD results when unpurified phage lysates were used. Thus,
This has been corrected. Now in lines 306 and 307.
Line 322: Thirdly,
This has been corrected. Now in Line 309.
Line 326 1-hour
This has been corrected. Now in Line 312.
Line 327: Finally, PMMD
This has been corrected. Now in Line 314.
Line 329: the words “great” and “very” are not needed in the sentence
The words have been eliminated.
Line 329-331 Phage K – sentence structure issues
“Phage K infects the great majority of S. aureus strains and only propagates in a very few isolates of very closely related Staphylococcus, such as S. hyicus, used sometimes as a surrogate for phage propagation for safety reasons [19,38]” has been changed to “Phage K infects the majority of S. aureus strains and only propagates in a few isolates of closely related Staphylococcus. This is the case of S. hyicus, used sometimes as a surrogate for phage K propagation for safety reasons [19,41].”
Please notice that the previous reference 38 is now reference 41. This was done to include three extra references, now references 37, 38 and 39, to answer a comment of reviewer 2.
Line 332: strains to >95%
This has been corrected. Now in line 318.
Line 332: In addition, due to the simplicity of PMMD, there
This has been corrected. Now in lines 318 and 319.
Line 335: as a novel, proof of concept method for the
This has been corrected. Now in line 321.
Line 336: using a S. aureus/phage K pair. PMMD shows promising characteristics
This has been corrected. Now in line 322.
Line 336-343: PMMD – sentence structure issues.
The original “PMMD shows very promising characteristics for its applicability for clinical and research diagnostic purposes, including sensitivity, specificity, cost, and speed. Although further development is required to cover most bacterial strains of a given species by using phage cocktails, because of the easiness to adapt this method quickly, it is a technology suitable for the rapid detection of specific bacterial strains causing epidemics as is. Moreover, due to the complete dependence on the specific phage to propagate in the bacterial strain of interest, this methodology could be adapted for the rapid selection of phages active against clinical isolates during phage therapy” have been replaced by “PMMD shows promising characteristics for its applicability for clinical and research diagnostic purposes. Nevertheless, further development is required to cover most bacterial strains of a given species, for example by using phage cocktails. Moreover, this methodology has the potential to be adapted for the rapid selection of phages active against clinical isolates during phage therapy”.
Reviewer 2 Report
Coupling the inherent discrimination of phage infection with the sensitivity and selection of PCR has been around a while, and although the current method utilises RTPCR detection of phage RNA, the principles are similar. I suggest the authors mention related methods, if only for the purpose of assay time and quantitative comparison. For example, Stanley et al 2007. Development of a new rapid combined phage and PCR method for detection and identification of viable Mycobacterium paratuberculosis bacteria within 48 hours. Appl. Environ. Microbiol. 73:1851-1857, but there are others.
The manuscript explores the metabolic differences of the target bacteria in the presence of the discriminatory antibiotic cefoxitin between MSSA and MRSA. Although interesting the impact of metabolic differences on the signals obtained should be investigated between target bacterial isolates, and the metabolic status of the bacteria recovered from challenging environmental and clinical samples.
The first paraph of the discussion is too general without any references, and much of it is already noted in the introduction.
Author Response
Thank you for your comments and suggestions. In the revised version of the manuscript we have tried to answer your points as well as we could.
Very Respectfully,
The authors.
Point by point responses:
Point 1
Coupling the inherent discrimination of phage infection with the sensitivity and selection of PCR has been around a while, and although the current method utilises RTPCR detection of phage RNA, the principles are similar. I suggest the authors mention related methods, if only for the purpose of assay time and quantitative comparison. For example, Stanley et al 2007. Development of a new rapid combined phage and PCR method for detection and identification of viable Mycobacterium paratuberculosis bacteria within 48 hours. Appl. Environ. Microbiol. 73:1851-1857, but there are others.
We have almost completely eliminated the first paragraph of the discussion, also in line with point 3, and instead we have incorporated a brief comment mentioning related methods using PCR. We have exemplified those methods by adding three extra references, now references 37, 38 and 39, including the suggested reference. In addition, we discuss about what we perceive as advantage of detecting phage RNA instead of genomic phage DNA, namely a shorter incubation time with the bacteria to produce signal above the background.
Point 2
The manuscript explores the metabolic differences of the target bacteria in the presence of the discriminatory antibiotic cefoxitin between MSSA and MRSA. Although interesting the impact of metabolic differences on the signals obtained should be investigated between target bacterial isolates, and the metabolic status of the bacteria recovered from challenging environmental and clinical samples.
Unfortunately, although an interesting question, currently we are not in a position to perform additional experiments on the metabolic status of the bacteria upon cefoxitin incubation. We believe though that our results showing a drastic decrease on phage RNA levels for the MSSA strain in the presence of cefoxitin and for the B. anthracis Sterne assay in presence of tetracycline, along with references 10, 16 and 25, are in line we our hypothesis. Nevertheless, we recognize that this matter requires further follow up studies.
Point 3
The first paraph of the discussion is too general without any references, and much of it is already noted in the introduction.
We agree that the first paragraph of the discussion is redundant and too general. We have almost completely deleted the paragraph and substituted by a more meaningful discussion, as indicated in the answer to point 1.
Round 2
Reviewer 2 Report
You should remove the eg from the references at line 324.
Author Response
The e.g. form the references 37, 38 and 39 of the first paragraph of the Discussion has been removed. Please notice that this is in line 288. Therefore the sentence has been changed from “Certainly, a number of previous studies have reported the successful use of phage PCR methods for bacterial detection [e.g., 37, 38, 39]” to “Certainly, a number of previous studies have reported the successful use of phage PCR methods for bacterial detection [37, 38, 39].”